# Patellar Resurfacing in Total Knee Arthroplasty, a Never-Ending Controversy; Case Report and Literature Review

**DOI:** 10.3390/diagnostics13030383

**Published:** 2023-01-19

**Authors:** Răzvan Adam, Cosmin Moldovan, Sorin Tudorache, Tudor Hârșovescu, Carmen Orban, Mark Pogărășteanu, Elena Rusu

**Affiliations:** 1Department of Orthopedics and Traumatology, Elias Emergency University Hospital, 011461 Bucharest, Romania; 2Department of First Aid and Disaster Medicine, Faculty of Medicine, Titu Maiorescu University of Bucharest, 040051 Bucharest, Romania; 3Department of Clinical Sciences, General Surgery, Faculty of Medicine, Titu Maiorescu University of Bucharest, 040051 Bucharest, Romania; 4Department of General Surgery, Witting Clinical Hospital, 010243 Bucharest, Romania; 5Department of Preclinical Sciences, Anatomy and Embryology, Faculty of Medicine, Titu Maiorescu University of Bucharest, 040051 Bucharest, Romania; 6Department of Anesthesia and Intensive Care, Carol Davila University of Medicine and Pharmacy, 020021 Bucharest, Romania; 7Intensive Care Unit Department, Monza Oncology Hospital, 013812 Bucharest, Romania; 8Department of Orthopedics and Traumatology, Carol Davila University of Medicine and Pharmacy, 020021 Bucharest, Romania; 9Department of Orthopedics and Traumatology, Dr. Carol Davila Central Military Emergency University Hospital, 010825 Bucharest, Romania; 10Department of Preclinical Sciences, Biochemistry, Faculty of Medicine, Titu Maiorescu University of Bucharest, 040051 Bucharest, Romania

**Keywords:** total knee replacement, patella, arthroplasty, resurfacing, anterior knee pain

## Abstract

Total knee arthroplasty (TKA) remains a lifesaving procedure for advanced gonarthrosis. However, patella resurfacing (PR) in TKA remains a controversial procedure, leading to extensive discussions amongst orthopedic surgeons, regarding its indications and results. Based on these premises, we present a clinical case of a 70-year-old Caucasian woman admitted for pain, swelling and limitation of left knee joint mobility. Her medical history records an Ahlback stage IV gonarthrosis with simultaneous bilateral TKA surgery performed in different hospital, when two NexGen cemented total prostheses were implanted with patellar resurfacing being performed only on the right side. Our clinical (American Knee Society Score, Lonner and Feller scales) and radiological evaluations (CT scan and Xray) revealed left patellar arthrosis and a slight lateral subluxation of the patella. The chosen treatment plan was revision surgery for PR and patellar prosthesis with a cemented patellar component, cross-linked polyethylene, no 32 NexGen model with 8.5 mm thickness. The immediate and distant postoperative evolution was favorable. Extensive literature review shows that, at present, PR remains at surgeon’s discretion mainly based on his previous results. Therefore, we believe there is an imperative need to develop high quality studies based on accurate scientific evidence to universally establish valid guidelines for PR in TKA.

## 1. Introduction

Total knee arthroplasty (TKA) remains a lifesaving procedure for advanced gonarthrosis. With the development of this procedure, both the operative techniques and the implants used have improved, leading to an increase in both quality of the surgical procedure and the patient’s life.

However, patella resurfacing in TKA remains a controversial procedure, leading to extensive discussions. In the early stages of knee replacement, especially because it is performed in advanced cases, patella resurfacing is routinely performed, leading to the disappearance of anterior knee pain (AKP) and improved joint function. Of course, alongside with the spreading of this technique came some complications of patellar fractures, aseptic necrosis, and patellar clunk syndrome (PCS).

These observations have led to the development of three major currents of opinion. One of those belong to the orthopedic surgeons that perform routine resurfacing because it will result in the disappearance of previous knee pain, improved joint function, decreased risk of reoperation, and remains a cost-effective procedure [1,2,3]. Another argument in favor of patellar resurfacing is given by the biomechanics of the patellofemoral joint. The pressure in these joints increases with knee flexion, with a maximum at 90–120 degrees.

Studies show that patellofemoral pressure is 5 to 7 times more of the body weight when standing up, 2 to 3 time more than the body weight when climbing stairs and 20 times more when jumping [4,5]. However, recent studies, using in vivo implanted sensors, show slightly lower values, but still high enough, such as 2 to 3 times body weight when getting up from a chair or climbing stairs, 15 times the body weight when jumping or 1 time the body weight when walking normally. These data may lead to the conclusion that degradation of articular cartilage and subchondral bone will progress even further, leading to anterior pain and discomfort when compared to nonresurfacing techniques.

Another group of surgeons consider that the complications mentioned above do not justify taking the risk of resurfacing. They argue that other procedures, such as electrocautery denervation, lateral retinaculum release or patelloplasty, can also lead to the disappearance of anterior knee pain [6,7].

Finally, there is a third current of opinion, namely surgeons who choose to selectively resurface the patella, depending on the degree of arthrosis or patella thickness, as the bone stock remaining after resurfacing is a serious problem.

Complications from patellar resurfacing can be caused by the surgical technique itself as well as the surgeon’s experience, the goal being to restore patellar thickness and its natural angle. The average patellar thickness in men is 25 mm and in women 22 mm [8]. The remaining bone thickness after resurfacing should be 12–15 mm [9]. Asymmetric patellar resurfacing technique is defined as a difference of more than 2 mm between the medial and lateral edges [10], which leads to the complications mentioned above.

Identifying the ideal resection plane is difficult. Many surgeons choose the free hand resurfacing technique, following anatomical landmarks and their own experience, repeatedly measuring lateral and medial thickness [11,12,13]. The use of cutting guides is, in theory, ideal but in practice they are difficult to apply and use. A study by Camp [14] compared the efficiency of free-hand resurfacing technique with the use of cutting guides and found that the free-hand technique produced more symmetrical patellar resections and thicknesses closer to the original than the cutting guide technique, mainly because of the difficulty of using them. The main risks associated with patella resection errors are shown in Table 1.

Based on these premises, we present a report case report that features a simultaneous bilateral total knee replacement with both nonresurfacing and resurfacing technique of the patella in the same patient, providing a unique opportunity to compare and assess, both from the patient’s point of view, and from clinical and imaging considerations, the effectiveness of this procedure. To better highlight how strong the debate of surfacing versus nonresurfacing is, we also conducted a literature review to objectify the clinical data of our case.

## 2. Case Report

We present the case of a 70-year-old Caucasian woman who was admitted in our orthopedic surgical department for pain, swelling and limitation of left knee joint mobility.

Four years ago, the patient was diagnosed with Ahlback stage IV gonarthrosis and underwent a simultaneous bilateral TKA surgery in another orthopedic clinic. Two NexGen cemented total prostheses were then implanted, with patellar resurfacing performed only on the right side, due to anesthetic complications which required a drastic shortening of the operating time, thus preventing the surgical team from performing resurfacing on the contralateral patella. Although postoperative radiological images showed a slight implantation error in the femoral component on the right side, extensive anterior resection with posterior rotation of the femoral component, the clinical outcome was satisfactory (Figure 1).

On the other hand, on the left side, where patellar resurfacing was not performed, although the radiological images show a satisfactory implantation of the prosthesis with correct alignment of the tibial and femoral components, but with slight subluxation of the patella, the clinical evolution was unsatisfactory (Figure 2). Shortly after operation, 3–4 weeks, the patient experienced discomfort, pain and swelling of the left knee.

Given the episodes of swelling associated with pain, an infectious inflammatory syndrome was initially suspected, for which intra-articular evacuation punctures were performed. Bacteriological examination of the specimen fluid was negative for pathological germs. The serum inflammatory markers also displayed normal values, as shown in Table 2 [15].

All these data refuted the infectious inflammatory etiology of the pain and swelling experienced by the patient. Upon being admitted in our clinic, the patient did not any signs of joint swelling but had typical complaints of anterior knee pain: pain in the anterior compartment around the patella, accentuated when getting up from the chair or descending steps.

The patient was clinically evaluated, initially using the American Knee Society Score (AKSS), but, as AKSS does not accurately assess patellofemoral function and anterior knee pain, we also used the Lonner and Feller patellofemoral scores in the clinical assessment. On the left knee, AKSS recorded a value of 28 points for the knee score and 30 points for the function score. The Lonner score counted 14 points in the pain assessment and 13 points in the function assessment, with a total of 27 points out of 100 maximum points of the scale. The Feller score was 6 points out 30 points.

On the right knee, where patellar resurfacing was performed, AKSS recorded a value of 88 points for the knee score and 100 points for the function score. Feller score recorded the maximum value of 30 points, and so did the Lonner score, with 100 points.

The difference between the clinical scores of the two knees before surgery is shown in Figure 3.

Alongside with the clinical scores, the patient was also evaluated radiologically, with CT scans and standard X-rays. Radiological investigations, using coronal, lateral and axial views (Merchant view) showed patellar arthrosis and a slight lateral subluxation of the patella (Figure 4).

CT images, subject to artifacts generated by the implant, show patellar wear and peripatellar inflammatory phenomena (Figure 5).

Based on these data, anterior knee pain secondary to patellar arthrosis and peripatellar tissue inflammation diagnosis was established. As such, the differential diagnosis of a septic joint process was ruled out.

The therapeutic method was decided, consisting of revision surgery for PR and patellar prosthesis. Since it is a reoperation and the risk of postoperative infection is higher than the primary intervention, we initiated prophylactic antibiotic treatment 2 days before surgery, based on i.v. broad-spectrum antibiotics (Cefuroxime 1.5 g every 12 h, and Ciprofloxacin 400 mg every 12 h).

During the procedure we detected advanced wear of the patella, the articular cartilage being completely absent exposing the subchondral bone, with osteocondensation process in place, in direct contact with the metal surface of the femoral component trochlea. Due to the advanced degree of wear of the patella, measuring its exact thickness was impossible, therefore we were forced to estimate it according to statistics; the set value was 22–23 mm. As such, we decided to implant a no. 32 NexGen model with cemented patellar component, cross-linked polyethylene, 8.5 mm thick. Thus, the resection was performed in such a way that the thickness of the remaining bone tissue to be precisely 13 mm, so that the total thickness of the patella would be approximately 22 mm.

The implant was placed close to the medial border of the resected patella, and a minimal lateral release was performed resulting in good patellar tracking in the femoral trochlea. At the same time, we removed the inflamed peripatellar tissues, completing the initial synovectomy, excising the peripatellar fibrosis and partially excising the inflamed part of Hoffa’s fat.

The immediate and distant postoperative evolution were favorable, as the radiological evaluation performed on the 2nd day after surgery shows; unfortunately, the lateral view is rotated (Figure 6).

The patient was mobilized early, with full loading, initiating active flexion/extension and isometric movements for the quadriceps muscle, starting from 2nd postoperative day.

Immediate postoperative pain was controlled with an epidural catheter and 10 mL ropivacaine 1% with fentanyl 2 mL (0.1 mg) and 38 mL of saline solution was injected using an automatic injector at rate of 5–7 mL/h. Nonsteroidal anti-inflammatory drugs (diclofenac) were also administered. This medication was used for two consecutive days. We also continued prophylactic antibiotic therapy for two more days after surgery.

Radiological and clinical follow-up were performed at 6 weeks, 3 months and then 12 months after surgery (Figure 7, Figure 8 and Figure 9).

The clinical evolution of the patient was also satisfactory. The anterior knee pain decreased significantly at 6 weeks after surgery and was absent at 3 months. At the same time, we recorded a significant improvement of the joint function, the episodes of swelling being completely absent. The aspect at 12 months after surgery is shown in Figure 10. The patient can easily achieve 120 degrees flexion without pain, full extension without joint swelling and is able at climbing stairs and get up from a chair, without pain.

Clinical evolution at 6 weeks and 3 months after surgery, was also monitored using the AKSS, Lonner and Feller scores, as shown in Table 3.

The overall clinical evolution of the left knee, reflected by the clinical scores, is shown in Figure 11.

Anterior knee pain (AKP), assessed by the ever increasing of Lonner pain scores, has decreased sharply, as shown in Figure 12.

## 3. Literature Review

Although our personal results of patellar resurfacing (PR) have been positive so far, this remains a controversial topic, and the literature is full of pro and con arguments regarding the widespread of this technique. For and objective comparison of our experience and in search of a possible international consensus, we conducted a descriptive review randomly selecting a total of 29 scientific publications, 17 papers being individual randomized trials of cohort studies from national arthroplasty registries, 9 meta-analysis and 3 literature reviews, published between 1984 and 2022 in PubMed, Scopus and Google Scholar databases. The search keywords were total knee replacement (TKA), patella replacement, patella resurfacing (PR), and anterior knee pain (AKP).

Abdulemir Ali [16] conducted a comparative study on a group of 74 patients with TKA, with a follow-up of 6 years. They were randomized into two groups, with and without PR. They were assessed using the VAS visual scale and KOOS (knee injury and osteoarthritis outcome score). He found no significant differences between the two groups in terms of postoperative pain or impaired joint function, concluding that PR is not a necessary intervention for a successful outcome of TKA.

Campbell et al. [17] reported, in another randomized study of 100 patients with gonarthrosis, who underwent TKA with or without PR. Clinical and radiological follow-up was 4 years and through questionnaire for up to 10 years. AKSS, Western Ontario and McMaster Universities Arthritis Index (WOMAC) were used for evaluation. A difference of 10 AKSS points between the two groups was considered statistically significant.

Their conclusion was that there were no statistically significant differences between the two groups, 65.8 points for the resurfacing group vs. 70.9, at 4 years respectively 53.5 points in the resurfacing group vs. 50.3 at 10 years.

Thus, although AKSS has no specificity for patellofemoral function and pain, the author does not recommend PR as a routine procedure in TKA.

Feller [18], in 1996, conducted an observational study on a group of 40 patients with TKA, who were randomized into two subgroups, with and without patellar resurfacing. The patients were evaluated using the HSS (Hospital for Special Surgery, New York, NY, USA) score and the Feller score, developed by the author himself, with a maximum of 30 points, a score that we also used in the evaluation of the clinical case presented. Follow up was at 6 months. The mean score at the end of follow-up was 89 HSS points and 28 Feller points in the unresurfaced group and 83 HSS points and 26 Feller points in the resurfaced group.

The author concludes that, although there were no complications in the patellar resurfacing group, there were no statistically significant differences between the two groups and does not consider that PR in TKA offers a statistically significant clinical benefit.

Etienne Deroche [19] conducted a prospective, randomized study on 250 operated knees, divided into two groups, with and without PR. All patients received the same type of prosthesis and were assessed using the International Knee Society Score (KSS), Forgotten Joint Score (FJS) and the Anterior Knee Pain Assessment (AKP). Their follow up was for 2 years.

The study concluded that there were no differences in prosthesis survival at 24 months between the two groups, resurfacing 88.3% vs. nonresurfacing 85.3% (*p* = 0.599). There were no statistically significant differences between KSS functional (*p* = 0.599), KSS knee (*p* = 0.396), FJS (*p* = 0.798) and AKP (*p* = 0.688), at 18 months follow up. In conclusion, routine PR is not recommended, as there are no significant differences between the two groups.

A large observational study in this direction was conducted by Stein Hakon Lastad Lygre [20]. The study analyzed 972 cases of TKA from the Norwegian National Arthroplasty Registry, recorded in the last two years prior to the study. Joint function and pain levels in cases with patellar resurfacing vs. nonresurfacing were assessed. Patients were divided into groups according to the prosthesis used AGC, Genesis I, or NexGen. 504 knees with resurfacing and 468 without PR were registered. They were assessed using the knee injury and osteoarthritis outcome score (KOOS) and the EQ 5D index score to evaluate the degree of satisfaction and the quality of health of each knee. The VAS scale was also used to assess pain levels.

The authors observed no statistically significant differences between the two groups, resurfaced and non-resurfaced, with estimated differences of <1.4 units and *p* value > 0.4. Thus, they consider that patellar resurfacing does not influence the clinical effect in terms of pain or function after TKA.

R.S.J. Burnett [21] conducted a prospective randomized study on a group of 32 patients, 64 knees who underwent bilateral TKA, using cruciate retaining (CR) prostheses. They were randomized into two subgroups, with and without PR, with a follow-up of 10 years. Patients were evaluated using the Knee Society Clinical Rating Score (KSCRS), a scale that ranges from 0 to 200 points maximum. The author observed no statistically significant differences in score between the two subgroups in terms of patient satisfaction, nonunion pain or revision rate. However, 2 patients, 7.4% of the nonresurfaced group and 1 patient, 3.5% of the resurfaced group required reoperation due to femuropatellar complications. He believes that the clinical results in TKA are similar with or without PR, a decision that remains at the surgeon’s discretion.

Se Jin Park [22] conducted a study on a group of 49 patients, 62 knees with TKA. They were divided into two groups with PR, 29 patients, 36 knees, follow up for 149 months and without PR, 20 patients, 26 knees, follow up for 140 months. They were clinically assessed using AKSS and HSS scores, as well as Lonner and Bristol scores to accurately assess patellofemoral function. No statistically significant differences were found between the two groups in terms of AKSS (95 points in the unresurfaced group vs. 93.5 points in the resurfaced group) and HSS (83 points in the unresurfaced group vs. 87 points in the resurfaced group). However, there were significant differences in the functional AKSS score, with 60 points in the unresurfaced group vs. 77.5 in the resurfaced group. Lonner and Bristol scores were similar in the 2 subgroups, 9 and 82 points respectively.

In conclusion, the author recommends that patellar resurfacing in TKA should be done only in selected cases. Although there were no significant overall differences in the scores evaluated, the functional score was significantly better in the case of resurfacing.

At the opposite end there are plenty of studies that demonstrate PR is a necessary procedure that can and should be used as a routine technique.

Akihide Kajino [23] performed 26 simultaneous bilateral TKAs in patients with rheumatoid arthritis, with PR randomly performed on one knee in each patient. In order to achieve good alignment, he performed lateral patellar release in all cases. Follow up was 6 years, and specific anterior knee pain symptoms associated with patellofemoral dysfunction were recorded in all cases without patellar resurfacing.

It concludes that, at least in patients with rheumatoid arthritis, PR in TKA is a necessary procedure.

Similar to these data, a study by Sledge and Ewald, [24] on a group of 1.474 TKA, with a 6 years follow up, suggested that failure to resurface the patella in rheumatoid arthritis will lead to the release of sequestered antigen from the remaining cartilage, resulting in recurrent joint inflammation. They recommend PR in all cases, especially those secondary to rheumatoid arthritis.

Similarly, Chengzhi Ha [25] conducted a study on a group of 66 patients with bilateral TKA, totaling 132 knees. They were randomized into two subgroups, with and without PR, and assessed using AKSS and Feller scores, with a follow-up of 5 years. Significant differences were recorded between the two groups, with increased AKSS and Feller scores (time effect *p* < 0.001), in the resurfaced vs. unresurfaced group, while no revisions were recorded in the resurfaced group. Thus, they conclude that PR is a necessary maneuver in TKA.

Another perspective on the evolution of the patellofemoral joint in the absence of PR in TKA is given by Dai Sato [26], in a study assessing the patellar status by MRI imaging. The study included 40 patients, 59 knees, evaluated over a 15-year period (2009–2014). Patients underwent TKA with zirconia ceramic femoral component without PR. They were evaluated radiologically and by the means of T2 weighted MRI, annually, with a follow-up of 5 years. The patellar cartilage was divided into 3 zones, medial, central, and lateral, for a better standardization of the evolution. Patients were clinically evaluated using AKSS and Japanese Oxford Score (JOS). It was determined that the mean lateral shift ratio increased significantly from 7.1% to 14.6%, and patellar cartilage thickness decreased significantly (*p* < 0.05), progressively decreasing in the three evaluated zones to less than half. This was reflected in the evolution of the clinical scores, recording low values at 5 years compared to one year after surgery. Four patients required reintervention for patellar prosthesis. Thus, the study concludes that patellar nonresurfacing in TKA will lead to significant decrease in patellar cartilage thickness, by more than 50% in 5 years, leading to femuropatellar dysfunction and requiring reintervention for patellar prosthesis.

The occurrence of anterior knee pain in the case of patellar nonresurfacing in TKA was also highlighted by Larry Rodrigues de Campos Junior in a cross-sectional study of 158 patients, totaling 162 knees undergoing TKA. These patients were randomized into two equal subgroups, with or without PR [27]. Clinical and functional outcome was assessed using Lequesne and WOMAC scores, with a follow-up of 5 years.

No significant differences in scores were observed between the two groups, Lequesne (*p* = 0.585) and WOMAC overall (*p* = 0.169) or in terms of subdivisions regarding stiffness, *p* = 0.796 and functional capacity, *p* = 0.196. However, a significant difference was noted in the WOMAC subdivision assessing pain, which was significantly lower in the resurfaced group, *p* = 0.036.

Thus, even if there are no statistically significant differences between the groups in terms of joint function, the nonresurfacing of the patella will still lead to anterior knee pain, affecting the patient’s quality of life.

David J. Wood, [28] conducted a prospective double-blind study of 220 knees undergoing TKA. They were randomized into two subgroups, 128 knees without PR and 92 with resurfacing, with a follow-up of 48 months. Patients were assessed using the Knee Society Clinical Rating Score (SCRS) and a specific assessment for anterior knee pain with a stair-climbing test.

Fifteen knees (12%) in the unresurfaced group and 9 (10%) in the resurfaced group required reoperation due to femuropatellar complications, the difference was not statistically significant (*p* = 0.650). However, the incidence of anterior knee pain was significantly higher in the unresurfaced group (*p* = 0.016). Thus, of the 128 knees that did not have patellar resurfacing, 39 (31%) had anterior knee pain and of the 92 with resurfacing, 15 (16%) had anterior knee pain at last follow-up. As such, the author considers that patellar nonresurfacing will lead to anterior knee pain, with impaired joint function and patient satisfaction.

These observations are also supported by another retrospective randomized study of Waters et al. [29]. They analyzed 514 TKA pressfit, divided into two subgroups, with and without PR. Follow up was 5.3 years, and patients were assessed using the KSS, AKP and British Orthopaedic Association Patient Satisfaction Score.

Anterior knee pain was recorded in 25.1% of cases in the unresurfaced group compared to 5.3% in the resurfaced group (*p* < 0.001). 11 patients required reintervention for patella prosthesis, of these 10 patients had no pain at all. The functional KSS score did not differ significantly between the two groups, but the KSS Knee and AKP scores were lower in the nonresurfaced group, the difference being statistically significant (*p* < 0.001). Thus, due to a significantly increased rate of pain in unresurfaced patella, they recommend resurfacing in TKA as a routine procedure.

The data from these individual studies are also confirmed by larger, cohort, researches such as the one by Joseph Coory [30]. He analyzed data from the Australian National Arthroplasty Registry on the differences between resurfacing and nonresurfacing patellar outcomes in TKA, performed between 1999 and 2017.

They analyzed 570.735 primary TKAs in which either post stability (PS) or cruciate retaining (CR) prosthesis types were used, with or without PR. Of these, 301.769 (52.9%) involved PR and 268.966 (47.1%) were unresurfaced procedures. There were 415,537 (72.8%) MS procedures of which 191.327 (46.0%) were MS patellar resurfaced and 224.210 (54.0%) were MS unresurfaced. There were 155.198 (27.2%) PS procedures of which 110.442 (71.2%) were PS patellar resurfaced and 44,756 (28.8%) were PS unresurfaced [30]. The study also analyzed the method of PR, in-lay or on-lay, and recorded the rate of revisions due to patellofemoral complications in the mentioned subgroups.

All primary TKAs where the patella was not resurfaced had a higher revision rate than those resurfaced (HR 1.31; CI: 1.28–1.35; *p* < 0.001). The values obtained in the selected groups are shown in the Table 4. In-lay resurfacing had a higher revision rate than on-lay (HR 1.27; CI: 1.17–1.37; *p* < 0.001). The conclusion of this study was that PR reduces the revision rate for both PS and CR prostheses, and that on-lay design is correlated with a lower revision rate.

Another large cohort study analyzing data from the Norwegian National Endoprosthetic Registry was conducted by Ove Furnes et al. [31]. He studied 7174 TKA registered between 1994 and 2000. Even though Nordic countries have a tendency to use nonresurfacing techniques for the patella [16,32,33], this study also recorded nonresurfacing prostheses in 65% of the cases. The 5-year survival rates of unresurfaced prostheses was 97%. In these cases, pain was the main symptom leading to surgical reintervention. The risk of revision due to pain was 5.7 times higher (*p* < 0.001) in unresurfaced prostheses compared to those with PR. Revision consisted of patella resurfacing. In resurfaced prostheses the main cause of revision was infection, with a 2.5-fold higher rate than in unresurfaced prostheses (*p* = 0.03), but a direct causal relationship with patella resurfacing could not be demonstrated.

Thus, according to the data obtained, nonresurfacing of the patella will lead to an increased risk of reintervention due to the occurrence of anterior knee pain.

In another cohort study, Alistair Maney [34] used data from the New Zeeland Joint Registry to analyze all three PR strategies, infrequent, selective, and frequent. Between 1999–2015, 57.766 primary TKAs performed by 203 surgeons were recorded. Resurfacing less than 10% of cases was defined as “rare”, between 10 and 90% as “selective” and >90% of cases as “frequent”. The revision rate and the Oxford Knee Score (OKS) at 6 months and 5 years postoperatively were evaluated.

Analyzed data revealed that 57% of surgeons resurface the patella selectively, 37% rarely and 7% frequently. The “frequent” group recorded the highest OKS values both at 6 months (38.57; *p* < 0.001) and 5 years postoperatively (41.34; *p* < 0.001), followed by the “selective” group (OKS—6 months 37.79, 5 years—40.87) and the “rare” group (OKS—6 months 36.92, 5 years—40.02). Overall, there were no significant differences in revision rates between the three groups (*p* = 0.587). In conclusion, although no significant differences in revision rates were recorded, routine PR resulted in improved patient-reported outcomes (Table 5).

For our review to be as objective as possible, we also analyzed data reported in meta-analyses and literature reviews focusing on patellar resurfacing in total knee replacement (TKA).

R.S. Nizard [35], in an attempt to systematize data from studies on patellar resurfacing, conducted a meta-analysis including 12 randomized trials reported between 1996 and 2003. The main outcomes analyzed were reoperation rates due to patellar problems, AKP, stair climbing ability, clinical score and patient satisfaction.

Randomized trials, considered the best design for comparing patellar resurfacing and nonresurfacing, were extracted from the Medline and Cochrane databases. We excluded studies with a follow up of less than one year, those in which the final follow up was performed by less than 80% of the subjects and those with less than 10 subjects included. The clinical scores were KSS and HSS. The 12 studies included 1.490 TKAs, 753 in the unresurfaced group and 737 in the resurfaced group, and 10 prosthesis designs were used.

The incidence of patellofemoral reoperation was significantly increased, 6.5% in the unresurfaced group compared to the resurfaced group, 2.3% (*p* = 0.0008). Incidence of AKP was significantly higher (*p* = 0.005) in the nonresurfaced group, with 22.3% vs. resurfaced, with 7.6%. A similar situation was observed for pain when climbing stairs, which was significantly higher in the nonresurfaced group (*p* = 0.01). In terms of patient satisfaction, the difference was not significant between the two groups and clinical scores could not provide concrete data due to the heterogeneity of the studies.

The author’s conclusion is that patellar resurfacing in TKA is an advantage, but due to the heterogeneity of the studies it is difficult to say whether it significantly influences functional outcome or patient satisfaction.

In another meta-analysis focused on the same topic, Alberto Delgado Gonzales [36] analyzed data from 11 randomized trials from PubMed, Scopus and Cochrane databases, published between 2010 and 2020. Clinical outcomes were represented by KSS, KOOS, Feller, WOMAC, VAS scores, and AKP. It also analyzed radiological outcomes, Insall–Salvati ratio, congruency angle or patellar tilt.

The KSS in knee assessment showed statistically significant differences in favor of the unresurfaced group (*p* = 0.0007), but the functional KSS did not show significant differences between the 2 groups. Similarly, the Feller score recorded a significant difference in favor of the resurfaced group (*p* = 0.001). The VAS scale was used in only three studies, the data analysis did not record significant differences.

The resurfaced group had a significantly higher overall risk of reoperation, *p* = 0.0137. The risk of AKP in the nonresurfaced group was significantly higher *p* = 0.012. In terms of radiological outcome, no significant differences were determined between the two groups.

The authors conclude that the evolution of clinical scores was better in the resurfaced group, which also had a lower rate of AKP and reoperation. However, he could not determine whether the increased risk was directly caused by femuropatellar complications, but it seems that resurfacing is a beneficial procedure within TKA.

Pacos Emilios [37] conducted a meta-analysis of randomized trials addressing patellar resurfacing in TKA, including 10 studies evaluating 1.223 knees. Main outcomes were revision rate, AKP occurrence rate after surgery and change in clinical KSS and HSS scores. Results showed that the absolute risk of revision was reduced by 4.6% in the resurfaced group (*p* < 0.001). PA reduced the risk of AKP by 13.8% (*p* < 0.01). Only four studies provided data on clinical scores, with data showing no significant differences between the two groups (*p* = 0.03).

Thus, the data obtained indicate that PA in TKA will reduce the occurrence of AKP and the risk of reoperation but will not significantly change the clinical outcome.

A meta-analysis by Piling R.W.D. [38] included 16 randomized trials that looked at PR, evaluating 3.465 TKAs, 1.710 with PA and 1755 without PA. The primary outcomes analyzed were KSS outcome, patient satisfaction and AKP occurrence and reoperation rates.

KSS knee recorded significantly higher values in the resurfaced group (*p* = 0.005), but functional KSS did not record significant differences. AKP was more frequent in the nonresurfaced group 24% vs. 13%, but the difference was not significant (*p* = 0.1). Patients were 90% satisfied in the resurfaced group vs. 89% in the unresurfaced group, the difference being not statistically significant. The reoperation rate due to AKP or other patellofemoral problems was significantly increased in the unresurfaced group (*p* < 0.00001 vs. *p* < 0.002).

Thus, although they recorded a higher reoperation rate in the unresurfaced group, the AKP rate and patient satisfaction levels did not differ significantly between the two groups, suggesting that a firm conclusion on patellar resurfacing in TKA cannot be drawn, leaving the decision solely on surgeon’s experience. This requires long follow-up studies focusing on the patellofemoral joint in TKA and using modern prosthesis designs.

A similar idea was also pursued by Umile G. Longo [2] in his meta-analysis of randomized, peer reviewed studies published in PubMed, Medline, Cochrane, CINAHL and EMBASE databases on patella resurfacing in TKA, the main outcome being clinical KSS, HSS scores and reoperation rate.

The KSS knee score was significantly increased in the resurfaced group (*p* = 0.004), but there were no significant differences in functional KSS. The HSS score recorded increased values in the resurfaced group compared to the unresurfaced group (*p* < 0.00001). The reoperation rate was significantly decreased in the resurfaced group, 1%, vs. unresurfaced 6.9% (*p* < 0.0001). In conclusion, based on the overall evolution of clinical scores, and the difference between the reoperation rate in the two groups, resurfacing the patella in TKA seems a more effective option. However, large, randomized studies with long-term follow-up are needed to reach an objective conclusion.

Yonghui Fu [39], conducted a meta-analysis of randomized trials addressing patellar resurfacing in TKA, including 10 studies that analyzed 1.003 TKAs. The main outcomes were revision rates, AKP occurrence rates and clinical score evolution. The absolute risk of revision was reduced by 4% in the resurfaced group (*p* = 0.06). In terms of AKP and clinical score there were no significant differences between the two groups. From the analysis of the data provided by the included studies, the author could not obtain a direct causal relationship between AKP and patella nonresurfacing. In conclusion, although PR reduces the risk of revision, but within a mild statistical difference, routine patella nonresurfacing is still a reasonable option.

In another meta-analysis, Filipo Migliorini [40] analyzed randomized trials focusing on PR in TKA, with outcomes such as AKP occurrence rate, revision rate, evolution of clinical KSS, HSS and range of motion scores. Studies were drawn from PubMed, Google Scholar, Scopus, and EMBASE databases. Only studies with evidence level I and II, with a minimum follow-up of 24 months, which provided quantitative data on the mentioned outcome, were accepted.

Thus, in the resurfaced group a significantly reduced rate of AKP occurrence, odd ratio 1.73, and revision, odd ratio 3.24, was determined. Clinical scores were higher in the resurfaced group KSS pain +0.97%, KSS clinical +0.23%, KSS functional +2.44%, KSS global +2.47%, HSS +5.64%. On the other hand, in the nonresurfaced group, increased ROM values were recorded, +3.09%. In conclusion, PR in TKA, overall, has superior performance, recording a reduced rate of AKP and revisions as well as increased values of clinical scores. However, from a functional point of view, the nonresurfaced group recorded a better ROM.

In terms of literature review, Alberto Grassi [41] conducted a review of meta-analyses from PubMed, Cinhal and Cochrane databases. He included in his review 10 meta-analyses published between 2005 and 2015, focusing on randomized trials of PR in TKA. Main outcomes were AKP occurrence rate, revision rate and functional score evolution. The quality of meta-analyses was checked with the AMSTAR score and the Jadad algorithm.

For functional scores, most meta-analyses did not identify significant differences between groups. AKP was significantly associated with the unresurfaced group in four meta-analyses while six did not register significant differences. In the case of revision, six meta-analyses identified a significantly increased risk in the unresurfaced group, with four describing the risk as related to patellofemoral complications. No study identified a low risk in the unresurfaced group. In conclusion, functional scores are not significantly altered by the patella resurfacing in TKA. The author considers that the increased rate of revisions in nonresurfacing should be interpreted with caution, due to the methodological limitations of meta-analyses in terms of search criteria and heterogeneity of studies, with patellar resurfacing indicated in certain well-selected cases.

Francesco Benazzo [42] conducted a review of studies in the literature addressing patellar resurfacing in TKA. He analyzed three main trends, always, never and selective resurfacing, considering the multitude of studies with controversial results. Geographically, there are different preferences for resurfacing. Thus, in North America, Australia or New Zealand resurfacing is overwhelmingly preferred, in about 90% of the cases. At the other end of the spectrum are Asian countries and northern European countries, Norway and Sweden, where the incidence rate is around 2–3% [43]. European countries generally prefer selective resurfacing, in about 50–70% of cases [44].

The author analyses resurfacing by implant-related issues, and, from the cited studies it appears that the CR cruciate retaining design is most often associated with nonresurfacing of the patella, however the upgrading of prostheses to the patella friendly type has led to a reduction of the resurfacing rate. Posterior stabilized PS implants are more often associated with patellar resurfacing.

Other issues were also statistically analyzed. This included surgical issues—some surgeons avoid resurfacing because cutting guides are imprecise and difficult to use. Most who choose resurfacing do it free hand, based on anatomical landmarks, chondro-osseous junction at the tendon insertion, and their own experiences. To optimize this, it is recommended that the remaining bone thickness be 12–15 mm, given that the average thickness of the patella in men is 26 mm and in women 23 mm. The transection should be symmetrical, with a maximum difference of 2 mm medial/lateral, usually with the medial edge being slightly thicker. It is also recommended to medialize the implant by about 2.5 mm and place it proximally to avoid PCS.

Patient related issues—studies show that there are 5 types of patients in whom resurfacing should be done. Patients with inflammatory diseases, obese patients, severe genu valgum, patients who need to climb stairs every day and women, the latter, according to the Swedish Knee Arthroplasty Registry [13,45], showing an increased degree of satisfaction after resurfacing. In conclusion, the author supports the idea of patellar resurfacing in TKA as a reproducible and safe procedure that reduces the risk of AKP and revisions.

As the optimal management of the patella in TKA remains a controversial topic, Kara McConaghy [46] conducted a review of randomized trials addressing this topic, the three main directions being routine resurfacing, nonresurfacing and selective resurfacing. As in Benazzo’s review she identifies the same geographical trends mentioned above. The author analyses the anatomy and biomechanics of the patellofemoral joint pointing out that the patella is subjected to pressures of 5 to 7 times the body weight when lifting from a chair, 2 to 3 times the body weight when climbing stairs and 20 times the body weight during jumps [47]. The management of patellar resurfacing is influenced by several factors. Resection technique plays an important role, and his observations are similar to those of Benazzo, with symmetry of the resection transection being the key factor in long-term outcome. Correct alignment and positioning of the implant is critical for correct patellar tracking in TKA [48], this influences patellofemoral stability and the occurrence of AKP, like the data in Benazzo’s study. In addition, they are also influenced by the rotation of the femoral and tibial components with excessive internal rotation having a negative effect on patellar tracking.

The design of the prosthesis, the material from which it is made and the type of fixation play an important role in determining the reoperation rate. With the development of highly cross-linked polyethylene, it is believed that the degree of wear will decrease and so will the rate of complications and reoperation. But the data in the literature are inconsistent, with some studies even recommending its use in TKA [49]. Cemented patella is favored by most authors with very good survival and outcome results [50]. Initially uncemented implants did not have good results but with the development of implants, they have recorded better and better results [51].

Regarding, the management of nonresurfacing, the method of patellar denervation was analyzed. This is most performed with electrocautery, with some studies showing decreased AKP after surgery [52]. However, in the long term, the method does not seem effective, and, according to some authors it is limited to 12 months after surgery [53]. Patelloplasty is a method consisting of patellar decompression, lateral patellectomy, osteophyte resection, patellar reshaping. The lack of standardization of these procedures makes it difficult to objectify them in studies. In conclusion, due to the abundance of contradictory data and the lack of clear standardization, in the author’s opinion, no clear conclusion can be drawn about which method is more effective in TKA.

Oliver Schindler [54] tries in his review to find a middle ground in the multitude of studies for reasons in favor and against PA. He analyses the conclusions and opinions of the authors of the studies in the literature, presenting a wide range of studies for and against, each with its own arguments.

Those supporting PA in TKA show a lower incidence of AKP, reoperation, complications and increased patient satisfaction and better joint function. Those who do not support resurfacing claim that the outcome differences mentioned above are not significant, resurfacing being an unnecessary surgical step. At the same time, nonresurfacing will lead to better patellar tracking with the ability to sustain higher patellofemoral forces and preservation of bone capital with a lower risk of osteonecrosis. It also considers the risk of implant wear, fracture, subluxations, associated with poor patellar implantation.

The selective resurfacing paradigm is to identify patients who will have an improved clinical outcome through resurfacing while avoiding potential complications associated with unnecessary resurfacing. In the case of patellar nonresurfacing, studies have shown changes in the shape and structure of the patella, which reshapes itself according to the shape of the femoral implant, while the bone becomes sclerotic. In terms of AKP, several studies show that nonresurfacing of the patella is more often associated with AKP, but other studies claim that AKP occurs in 10% of TKAs whether the patella is resurfaced [48]. Several predictors for AKP after surgery have been suggested, but only a few, such as obesity or flexion contracture have been objectified, with most studies being unable to establish a clear difference between AKP-affected and unaffected TKA. The analysis of data from different national arthroplasty registries confirmed the geographical distribution of patellar resurfacing tendencies in TKA, as mentioned in the other reviews above.

The author’s conclusion is that the patella is an essential component with a major role in TKA survival, and its correct management plays a very important role. Unfortunately, current studies and meta-analyses cannot indicate a concrete direction supported by solid scientific arguments, without being able to provide surgeons with specific guidance. Thus, selective resurfacing seems a tempting proposal, the important thing being the correct selection of patients. However, the selection criteria remain elusive, based only on intuitive reasons. It is therefore important to develop correct indicators to identify which patients can benefit from patella resurfacing and which cannot. In the midst of this battle of pros and cons, the author believes that a compromise solution such as selective resurfacing may be the right choice for the majority of patients.

Shuzhen Li [55] conducted a review of randomized trials of patellar resurfacing in TKA, and then did a meta-analysis of the data obtained. The studies were extracted from the Cochrane, Medline and Embase databases and included 14 studies, 1603 TKA, 817 without patellar resurfacing and 786 with resurfacing and follow up between 1 to 10 years. Primary outcomes were the reoperation rates and AKP. Reoperation was subdivided into two categories, first, patellofemoral complications and second patellofemoral complications, infections, periprosthetic fractures. Secondary outcomes were patient’s satisfaction and evolution of main clinical scores, such as KSS, HSS, Bristol Knee Score. The overall reoperation rate was 3.9% in the resurfaced group and 7.8% in the unresurfaced group, with random effects statistical analysis indicating a *p* = 0.06, results not reliable. Subgroup analysis at a follow-up of less than 5 years showed no significant differences between groups. However, at a follow-up of more than 5 years the difference was statistically significant. The relative risk in the resurfacing group was 0.27 times lower than in the other group. The absolute risk was reduced by 0.08 in the resurfaced group. The reoperation rate for femoral patellar causes in the resurfaced group was significantly decreased compared to the nonresurfaced group, 1.8% vs. 6.2% (*p* = 0.0001). The risk of AKP was significantly increased in the nonresurfaced group, 24.1% vs. resurfaced, 12.9% (*p* < 0.00001). However, substantial heterogeneity was detected across these studies. When the analysis was restricted to the highest quality studies, no significant difference between groups was detected.

Clinical scores did not show major differences between groups, regardless of the analysis method used. The same was true for patient satisfaction (*p* = 0.63).

Based on the data obtained, the author concluded that PR in TKA reduces the risk of reoperation, especially on the long term. However, it does not significantly influence joint function, AKP rate, or patient satisfaction. Thus, although PR appears to be an effective and safe method, routine resurfacing is not supported by sufficient evidence, which favors selective resurfacing, with some selection criteria based on the quality of the patellofemoral joint, the surgeon’s experience or the design of the prosthesis used.

The main data from the above meta-analysis and reviews are presented in Table 6.

## 4. Discussion

Patella is an important component in TKA, its proper management playing an important role not only in patient’s satisfaction or joint function but also in the survival of TKA itself.

Our case report is interesting and unique because it features a simultaneous bilateral total knee replacement with both nonresurfacing and resurfacing technique of the patella in the same patient, providing a unique opportunity to compare and assess, both from the patient’s point of view, and from clinical and imaging considerations, the effectiveness of this procedure. Even though not all orthopedic surgeons agree with prophylactic antibiotherapy prior to surgery, we decided to take all necessary measures to rule out any infectious postoperative complications or even sepsis, though a remote complication but still possible, especially in reinterventions [56]. Although PR was initially intended to be performed bilaterally, because of anesthesia-related reasons, this procedure was done only on the left side, and it was not part of the surgeon’s original strategy. The prosthesis on the right side, where the patella was resurfaced, although presenting small implantation imperfections, had a satisfactory clinical evolution, while the prosthesis on the left side, correctly implanted, but where the patella was not resurfaced, had an unfavorable clinical evolution. Four years after surgery, the left-sided prosthesis shows clear signs of anterior knee pain, reduced clinical scores, especially those that assess the patellofemoral joint, such as Feller and Lonner, and, at the same time, clear radiological signs of patellar wear, fibrosis and peripatellar inflamed tissue production.

Although many studies in the literature claim that there are no significant differences between the functional scores, 4 years after the initial arthroplasty the patient showed a large difference between the AKSS and Lonner functional scores of the resurfaced knee compared to the unresurfaced one. In our case, PA surgery resulted in improvement of the above clinical scores and overall joint function, at 12 months follow up after surgery, reaching maximum values.

Patellar resurfacing remains a controversial topic that divides the orthopedic surgical community, with each side bringing arguments proven in a multitude of studies.

Studies that do not recommend patella resurfacing in TKA are based on the lack of a statistically significant difference between the two groups analyzed in terms of AKP, revision rate or clinical score evolution. However, in this type of studies, there are no values of the mentioned parameters that are significantly favorable to the nonresurfaced group. In contrast, studies supporting patella resurfacing show significantly improved values of AKP, revision rate or clinical scores in the resurfaced group.

Follow-up plays an important role in determining the degree of objectivity of a study. A too short follow-up, usually less than 5 years, probably does not allow objective observation of patellofemoral complications. As observed in Shuzhen’s meta-analysis [55], a significant difference in revision rate only appears when the follow-up was extended beyond 5 years mark.

The average follow-up of the five studies not recommending patellar resurfacing in TKA is 4.1 years, with a range spanning from 0.5 to 10 years. The mean follow-up of the two studies recommending selective resurfacing was 11 years (range 10 to 12 years), and of the 10 studies recommending resurfacing the median was 6.33 years (range 4–17 years). As such, the higher the follow-up interval, the higher the incidence of complications in the nonresurfaced group, explained by the wear of the patella and extensor apparatus progressively sets in over time. This progression of patellar wear is also objectified in Dai Sato’s MRI study [26] which showed a major decrease in patellar cartilage thickness and clinical scores between 1 and 5 years after surgery.

The average number of patients included in the 5 studies not recommending resurfacing is 287.2, including a cohort study from the Norwegian National Arthroplasty Registry, [20] which included 972 cases. The 2 studies recommending selective resurfacing have an average of 63 cases and the 10 studies recommending patellar resurfacing have an average of 63,826.2 cases. But this group included 3 cohort studies [30,31,34] from the Australian, New Zealand and Norwegian national registries. If we exclude the cohort studies, the unresurfaced group has an average of cases and the resurfaced group 369.57 cases. Regardless of how we choose to analyze these cases, studies favoring patellar resurfacing in TKA are more consistent by including a larger number of cases, which gives them a degree of objectivity and increased scientific value. Cohort studies in national registries are very valuable, analyzing many cases registered over long periods of time, 17–19 years. Their weakness is that they are influenced by geographical trends, as evidenced by the meta-analysis and review data included in our study. Thus Australia, New Zealand along with North America tend to resurface 90% of cases. While countries in Asia or Northern Europe, Norway, Sweden resurface in a very small percentage, 2–3% of cases [42,46]. Interestingly, in our random database selection, we included a cohort study [31] from the Norwegian national arthroplasty registry, which favors patellar prosthesis in TKA.

Of the 9 meta-analyses included in our descriptive review, 5 recommend PA, one does not, and 3 recommend selective resurfacing. All three reviews recommend selective resurfacing, but one still finds it difficult for a concrete conclusion to be reached at this time due to the multitude of conflicting data in studies addressing this topic.

The main outcomes analyzed in the meta-analyses were AKP, reoperation rate, stair climbing ability, patient satisfaction and clinical scores. The vast majority only analyzed KSS and HSS scores, which do not have high specificity for the patellofemoral joint. In the case of meta-analyses recommending patellar resurfacing, a low AKP and reoperation rate and increased values of stair climbing ability and pain scores were observed in the resurfaced group. In contrast, functional scores or patient satisfaction do not show statistically significant differences. The Feller score, specific for patellofemoral disorders, has been used quite rarely, but has registered significant differences for the resurfaced group, as Alberto Delgado Gonzales shows [36].

Thus, although most meta-analyses consider PA in TKA as an advantage, due to the heterogeneity of studies it is difficult to say whether it significantly influences the patient’s functional outcome.

Meta-analysis not supporting resurfacing recorded insignificant differences in AKP and clinical score between the two groups and a mild, insignificant increase in the reoperation rate.

Meta-analyses recommending selective resurfacing did not record significant differences in AKP and clinical score between groups, but the reoperation rate was significantly increased in the nonresurfaced group, especially with follow ups beyond 5 years.

As the literature reviews show, in all this battle of pros and cons, selective resurfacing of the patella in TKA may be a fair compromise solution [54]. It may increase patient satisfaction, but the big disadvantage remains correct patient selection. Currently, the selection criteria remain unclear, subjective, and based mainly on the surgeon’s experience and intuition.

The role of the design of the femoral component of the prosthesis, the material of the patellar implant and its fixation are also widely debated in literature reviews. The design of the femoral component plays an important role in the evolution of the patellofemoral joint, especially in the case of nonresurfacing. Even though the studies included in our review prompted that most prostheses had a modern design, with anatomically shaped femoral trochlea, adapted to the natural patella, data showed a predominance of AKP and reoperation rates in the unresurfaced group.

Another important role in the evolution of the patellofemoral joint is patella resection and obtaining a symmetrical transection with maximum 2 mm difference between the medial and lateral edges, this goal being essential for a good outcome. Likewise, implantation following certain landmarks, medialization, and proximal placement of the patellar implant, can lead to correct patellar tracking.

Regarding patellar nonresurfacing, the most common procedures are electrocautery denervation, osteophyte resection and patelloplasty. But, as mentioned in Kara McConaghy’s review [46], the effect of denervation diminishes markedly after 12 months.

## 5. Conclusions

PA in TKA remains a widely debated topic, the most critical aspect of this debate being the emergence of AKP, the evolution of functional scores and the reoperation rates. In the case of the latter, reoperation due to AKP and patellar dysfunction is weighed against reoperation due to loosening and patellar fracture.

AKP occurrence and reoperation rates are more often associated with TKA without patella resurfacing, as seen in the cited literature and in our reported case. Clinical scores assessing pain have favorable values in the resurfaced group. Although many studies claim that there are no significant differences in functional scores, however, long follow up evaluations, over 5 years, have shown differences in favor of the resurfaced group, as we observed in the reported case.

Patella lesions are slow and with a progressive onset, requiring a longer period, 4–5 years, to become clinically visible.

As in our reported case, studies with follow up over 5 years have noted patellar wear, with subchondral bone structure changes and patellar deformities. This evolution induced by the contact of a cartilage still in a degraded state with the metal of the femoral component, even though fitted with a modern design, seems similar to the changes induced by the cotiloiditis secondary to hemiarthroplasty with Austin–Moore prosthesis, which is also manifested by pain and limitation of joint function.

The randomized trials provide conflicting but scientifically supported arguments for both, pro- and against PR. However, studies supporting resurfacing, overall, have a longer follow-up and a higher number of cases included.

Studies from national registries, analyzing many cases, provide important scientific input but are influenced by geographical trends. However, we have observed that, overall, the global values tend toward resurfacing of the patella in TKA.

The same trend is observed in meta-analyses and reviews, but they recognize that the heterogeneity of studies is an obstacle to obtaining an objective analysis. However, the vast majority of those randomly selected in our review recommend routine or selective patellar resurfacing in TKA.

Following multiple contrasting data in the literature, selective patellar resurfacing seems a reasonable compromise solution. However, it is limited by the lack of objective patient selection criteria. Until these are established, patellar resurfacing will remain a compromise solution.

There is an imperative need to develop high quality studies based on accurate scientific evidence to establish a universally valid guideline for patella resurfacing in TKA.

At present, patellar resurfacing remains at the discretion of the orthopedic surgeon being influenced by the surgeon’s own results.

Thus, based on our personal results, the clinical case presented that brings additional scientific arguments, and on the conclusions obtained from the analysis of the literature data, our option inclines toward routine resurfacing of the patella in TKA.

## Figures and Tables

**Figure 1 diagnostics-13-00383-f001:**
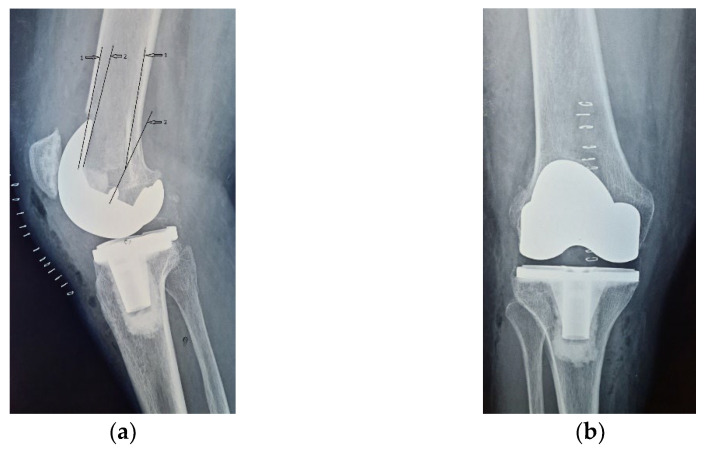
Right knee, initial after surgery radiological image: (**a**) lateral view, extensive anterior resection, femoral posterior rotation, femoral component axis intersecting diaphyseal axis at a 15.4° angle with (1) marking the femoral diaphyseal axis and (2) marking femoral component rotation axis; (**b**) coronal view.

**Figure 2 diagnostics-13-00383-f002:**
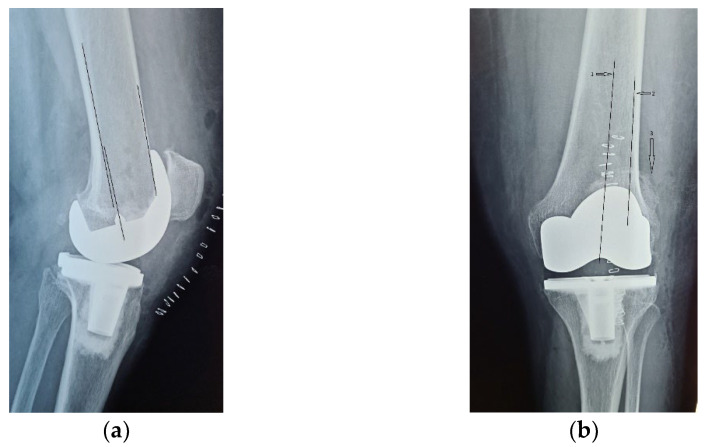
Initial radiological aspect of the left knee, right after surgery: (**a**) lateral view, femoral component axis parallel with femoral diaphyseal axis; (**b**) coronal view, slight lateral patellar subluxation (arrow) compared to femoral axis with (1) marking the femoral trochlear axis, (2) the patellar axis, and (3) patella must be centered in the femoral trochlea.

**Figure 3 diagnostics-13-00383-f003:**
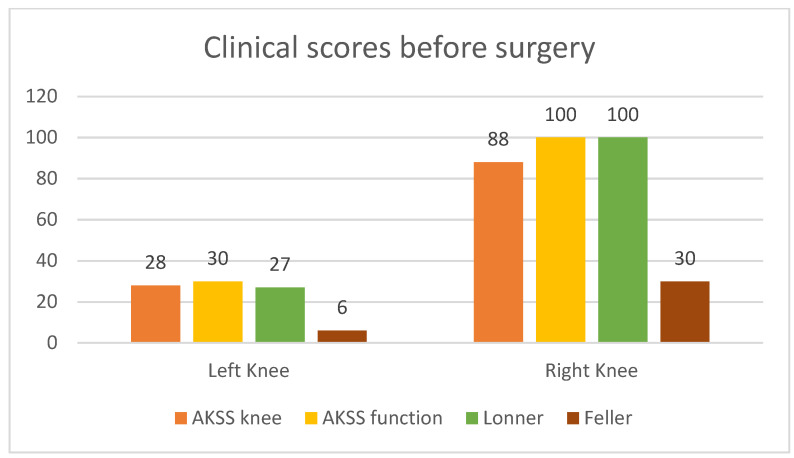
Comparison between clinical scores, before surgery.

**Figure 4 diagnostics-13-00383-f004:**
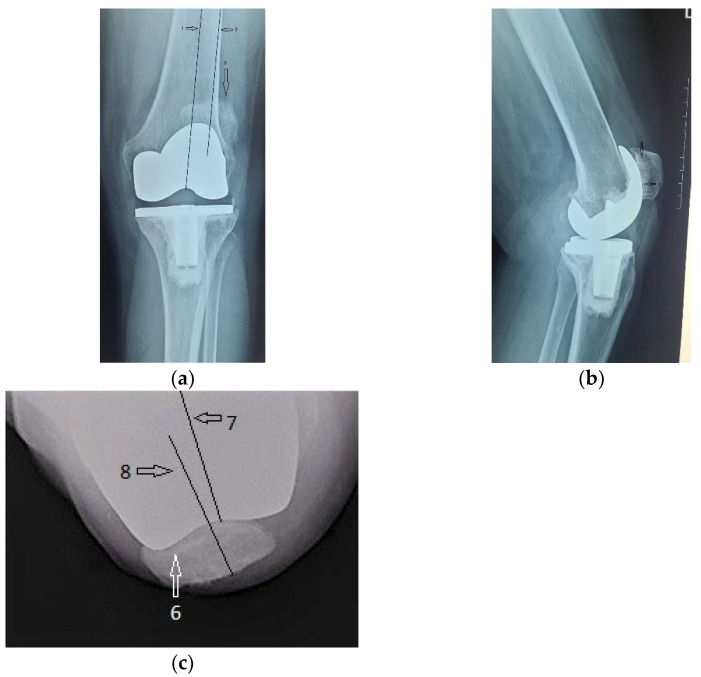
(**a**) coronal view, patellar lateral subluxation (arrow) compared to femoral axis with (1) marking the femoral trochlear axis, (2) patellar axis and (3) the patella must be centered in the femoral trochlea; (**b**) lateral view, patellar wear, patellar trochlea deformation and subchondral bone condensation with (4) marking subchondral osteocondensation and (5) patella bone deformation; (**c**) axial view, patellar lateral subluxation, and deformation, osteocondesation of the subchondral bone with (6) patella bone deformation and condensation, (7) femoral trochlear axis and (8) Patellar axis.

**Figure 5 diagnostics-13-00383-f005:**
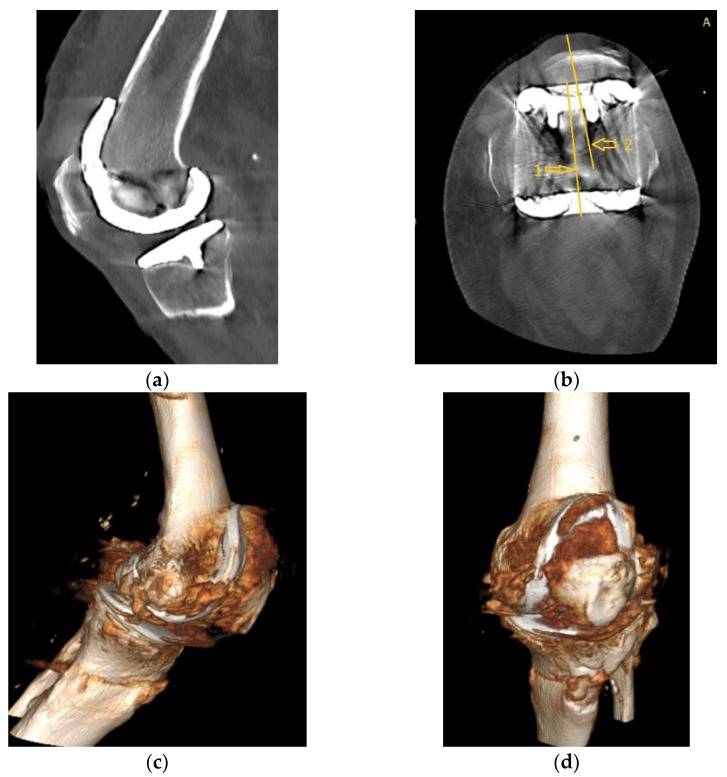
CT images: (**a**) Sagittal view showing patellar arthrosis, condensation of subchondral bone. (**b**) Axial view showing patellar arthrosis, deformation, and lateral subluxation (arrow) with (1) marking the femoral trochlear axis and (2) patellar axis; (**c**,**d**) CT 3D image, peripatellar inflammation.

**Figure 6 diagnostics-13-00383-f006:**
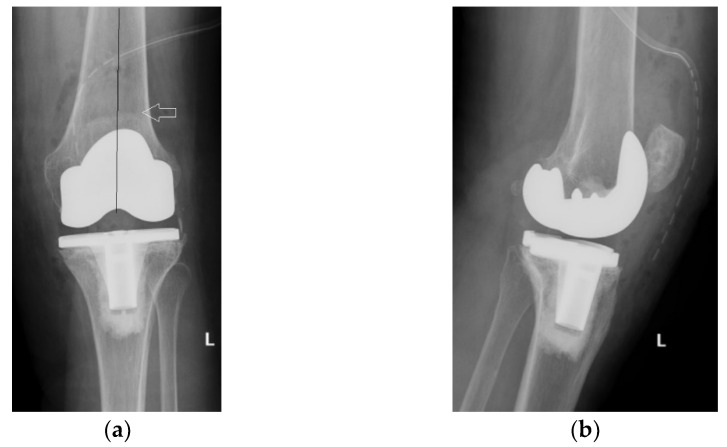
(**a**) Coronal view, patella centered on femoral axis (arrow); (**b**) Lateral view.

**Figure 7 diagnostics-13-00383-f007:**
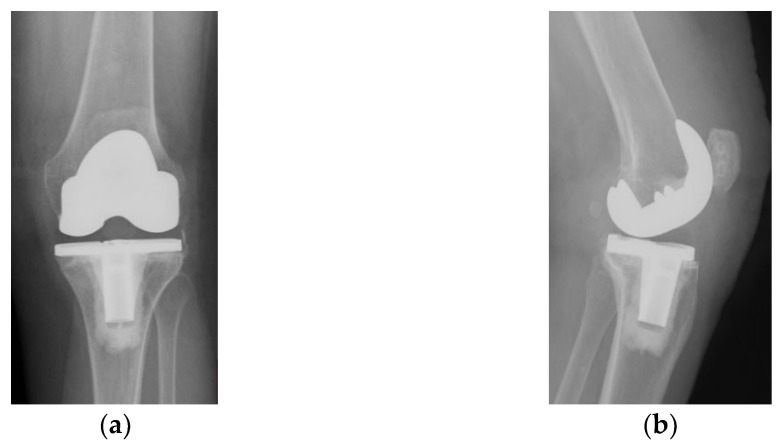
Radiological imaging at 6 weeks after surgery: (**a**) Coronal view; (**b**) Lateral view.

**Figure 8 diagnostics-13-00383-f008:**
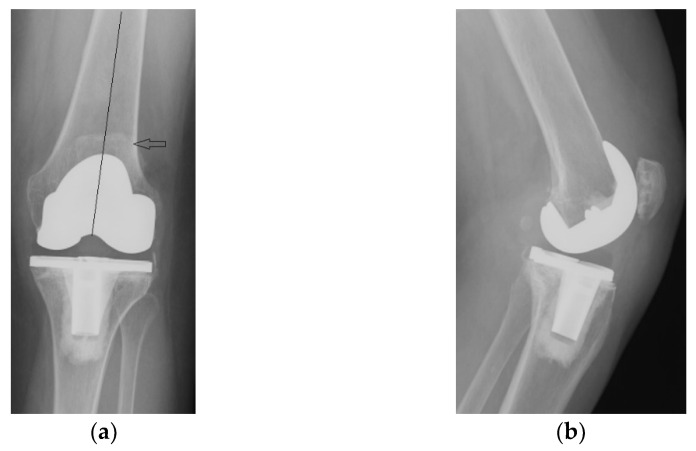
Radiological imaging at 3 months after surgery. (**a**) Coronal view, patella is centered to the femoral component (arrow); (**b**) lateral view.

**Figure 9 diagnostics-13-00383-f009:**
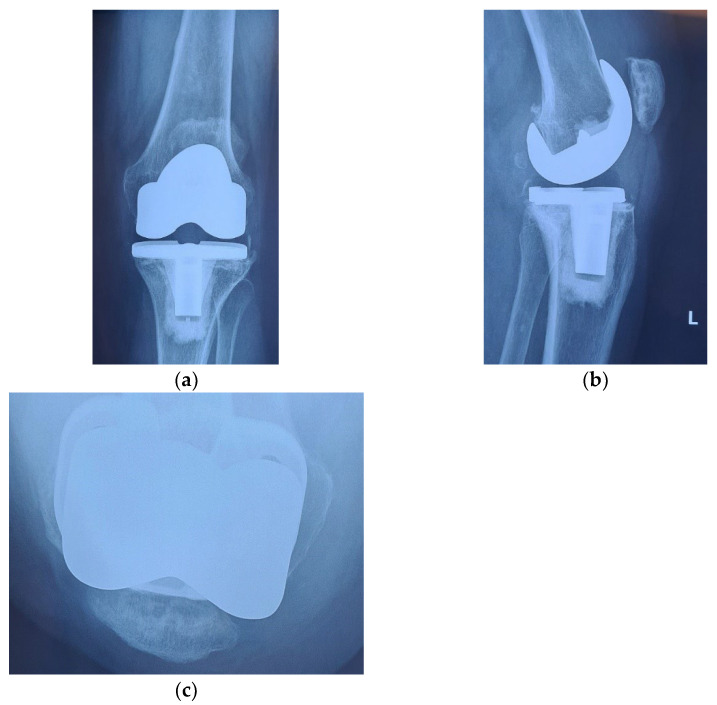
12 months after surgery: (**a**) coronal view, normal centered patella; (**b**) lateral view, no signs of patellar loosening; (**c**) axial view, normal centered patella in femoral trochlea.

**Figure 10 diagnostics-13-00383-f010:**
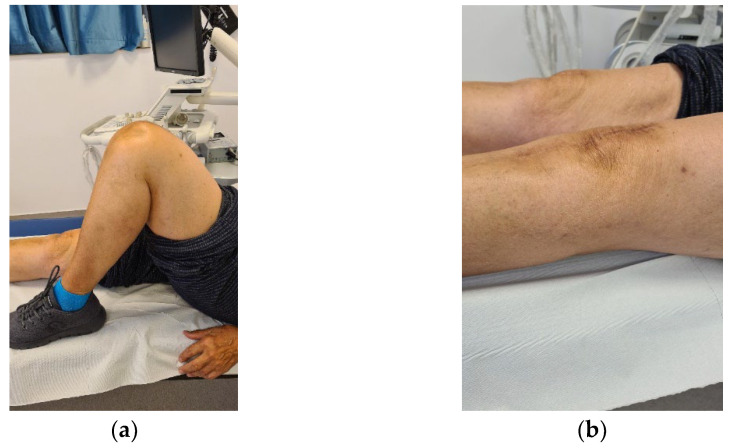
Clinical aspect at 12 months after surgery; (**a**) 120-degree flexion; (**b**). Complete extension and no join swelling.

**Figure 11 diagnostics-13-00383-f011:**
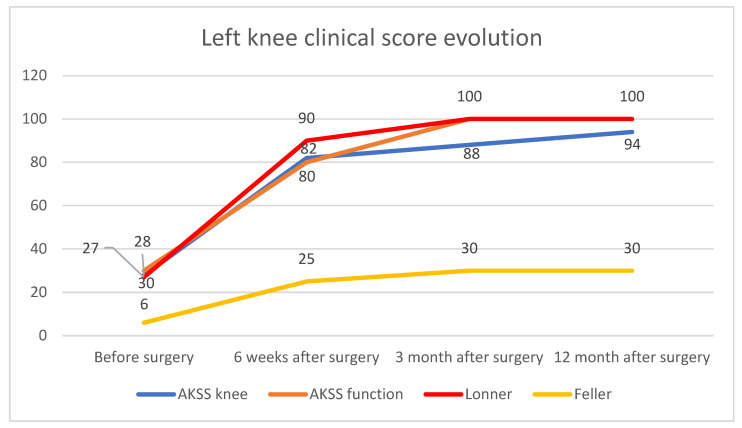
Overall left knee clinical score evolution.

**Figure 12 diagnostics-13-00383-f012:**
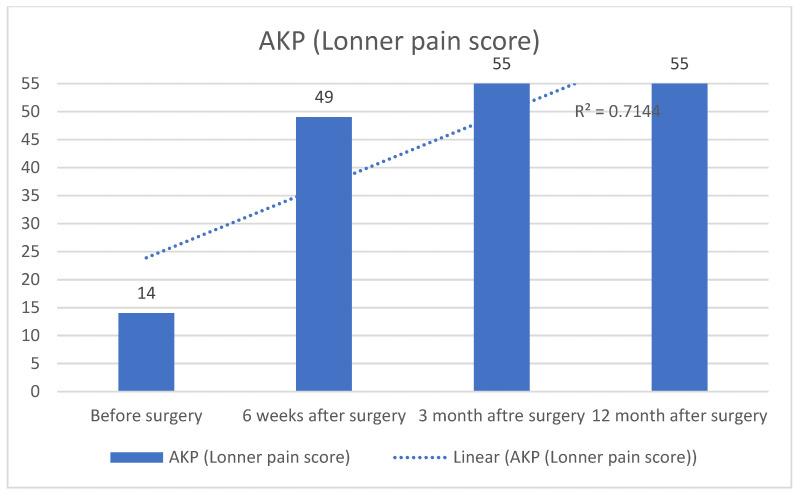
Lonner pain score evolution.

**Table 1 diagnostics-13-00383-t001:** Risks associated with patella resection errors (adapted after [1]).

Limited Resection	Extensive Resection
Maltracking and increased wear	Low mechanical strength with risk of fracture
Limitation of joint mobility	Loosening
Increased patellofemoral pressure	Patellar clunk syndrome
Impairment of quadriceps biomechanics	Impairment of extensor mechanism
Increased postoperative pain	Increased strain on the anterior

**Table 2 diagnostics-13-00383-t002:** Inflammatory panel of the patient.

Markers	Registered Values	Normal Values/Rage
C Reactive Protein (CRP)	0.97 mg/L	<5 mg/L
Erythrocyte Sedimentation Rate (ESR)	2 mm/h	<30 mm/h
Fibrinogen	309 mg/dL	200–400 mg/dL
Leukocyte count	5.88 × 10^3^/µL	4–10 × 10^3^/µL

**Table 3 diagnostics-13-00383-t003:** Comparison between clinical scores and their evolution.

Clinical Scoring System	Before Surgery		6 Weeks after Surgery		3 Months after Surgery		12 Months after Surgery	
AKSS (Knee/Function)	28	30	82	80	88	100	94	100
Lonner (Total/Pain/Function)	27		90		100		100	
	14	13	49	41	55	45	55	55
Feller	6		25		30		30	

**Table 4 diagnostics-13-00383-t004:** Revision rates, by groups [30].

	PS Unresurfaced	MS Unresurfaced	PS Resurfaced	MS Resurfaced
Total revision rate at 17 years	11.1%	8.8%	7.9%	7.1%
Direct patellar related revision	2.74%	1.60%	0.25%	0.25%

**Table 5 diagnostics-13-00383-t005:** Data from the main PR studies.

Author	No. of Cases	Follow Up	Recommend Resurfacing	Not Recommending Resurfacing	Selective Resurfacing	Significant Difference between Groups
Abdulemir Ali	74	6 years	-	+	-	no
Campbell	100	4–10 years	-	+	-	no
Feller	40	6 months	-	+	-	no
Etienne Deroche	250	2 years	-	+	-	no
Stein Hakon Lastad Lygre	972	2 years NR ^1^	-	+	-	no
R.S.J. Burnett	64	10 years	-	-	+	no
Se Jin Park	62	11–12 years	-	-	+	no
Akihide Kajino	26	6 years	+	-	-	yes
Sledge and Ewald	1474	6 years	+	-	-	noncomparative
Chengzhi Ha	132	5 years	+	-	-	yes
Dai Sato	59	5 years	+	-	-	yes
Larry Rodrigues de Campos	162	5 years	+	-	-	yes
David J. Wood	220	4 years	+	-	-	yes

^1^ NR: National Registry.

**Table 6 diagnostics-13-00383-t006:** Main data for PR meta-analysis and reviews.

Author	Type	No. of Studies/TKA	Statistically Significant Difference	Recommend Resurfacing	Do Not Recommend Resurfacing	Selective Resurfacing
R.S. Nizard	Meta-analysis	12 1490 TKAs	Yes to AKP, revision, No to CS	+	-	-
Alberto Delgado Gonzales	Meta-analysis	11	Yes to AKP, revision, CS	+	-	-
Pacos Emilios	Meta-analysis	10 1223 TKAs	Yes to AKP, revision, No to CS	+	-	-
Piling R.W.D.	Meta-analysis	16 3465 TKAs	No to AKP, CS Yes to revision	-	-	+
Umile G. Longo	Meta-analysis	11	Yes to revision, CS	+	-	-
Yonghui Fu	Meta-analysis	10 1003 TKAs	No to AKP, CS Mild to revision	-	+	-
Filipo Migliorini	Meta-analysis	12	Yes to AKP, revision, CS	+	-	-
Alberto Grassi	Meta-analysis	10	No to AKP, CS Yes to revision	-	-	+
Shuzhen Li	Meta-analysis review	14 1603 TKA	No to AKP, CS Yes to revision long term	-	-	+
Francesc Benazzo	Review	-	-	-	-	+
Kara McConaghy	Review	-	-	-	-	+/-
Oliver Schindler	Review	-	-	-	-	+

## Data Availability

All clinical and imaging data of the patient are available upon written request from the main or corresponding authors.

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
