# Peer review of "Patellar Resurfacing in Total Knee Arthroplasty, a Never-Ending Controversy; Case Report and Literature Review"

_diagnostics, 2023, doi:10.3390/diagnostics13030383_

Round 1
Reviewer 1 Report
• The difference and innovative aspects of the study from the literature should be emphasized at the end of the introduction.
• In Figure 1, the implantation error should be pointed out more clearly.
• In Figure 2, the subluxation region should be marked more clearly.
• “Bacteriological examination of the specimen fluid was negative for pathological germs.” If the relevant results are ready for use, the study can be enriched by including them in the study.
• “The serum inflammatory markers also displayed normal values, as shown in table 2.” References can be made from the literature to support the sentence.
• The values in Figure 3. can be compared with the literature.
• Separate figures in all figures must be aligned.
• In Figure 4, the subluxation region should be pointed out more clearly. Similarly, other Figures should be edited.
• mL should be used instead of ml. dL should be used instead of dl. Similarly, units should be checked.
• The values in Figure 11. can be compared with the literature.
• The values in Figure 12. can be compared with the literature.
• The contribution of the study to the literature should be emphasized in the conclusion part.
• References for recent years can be increased (especially in 2022 and 2023).
Author Response
All modifications requested by the reviewers have been marked in the revised manuscript with "track changes" function in Microsoft Word (please see attachment).
(1) "The difference and innovative aspects of the study from the literature should be emphasized at the end of the introduction" - As instructed, we revised the importance of the study in the last part of the INTRODUCTION section.
(2) Figure 1 a has been modified to better highlight the implantation error. As such, new labels (numbers) have been added to reflect more clearly the content.
(3) As requested by Reviewer 1, the image was modified to better reflect the subluxation region. As such, new labels with numbers have been added.
(4) The relevant results are already available and were all negative for any type of bacterial contamination (sterile).
(5) Although we believe that normal values for biological data should not be cited, a new reference was added to better highlight the values in Table 2 - citation index [15].
(6) The values depicted in figure 3 represent a comparison between the pre- and postoperative states of the patient, this data does not require comparison with the published data.
(7) "Separate figures in all figures must be aligned" - please be more specific. For image insertion we used the provided table layout in the official MDPI template for Diagnostics. It automatically allignes the inserted pictures.
(8) As requested by Reviewer 1, the images in figure 4 and figure 5 have been modified to better reflect the description in text. Also, new labels have been added.
(9) "mL should be used instead of ml. dL should be used instead of dl. Similarly, units should be checked." - we checked and revised all units, as instructed.
(10) + (11) This is a unique case of a patient operated with 2 different procedures in two consecutive surgeries on different knees by different surgical teams. In our opinion comparing the values before and after surgery for each individual procedure with published data does not fit into the scopes of this paper.
(12) "The contribution of the study to the literature should be emphasized in the conclusion part" - we changed the conclusions to better reflect the contributions.
(13) "References for recent years can be increased (especially in 2022 and 2023)." - The reference list does contain papers published in 2022. The citation of all papers in 2022 was not made because our selection of scientific materials has been made on a random basis so that we would not be biased towards one method or the other. To hour knowledge, no papers in 2023 have been published so far.

Reviewer 2 Report
The article presented by authors and, in particular, the clinical case described by them, are of interest to specialists in the field of joint replacement. At the same time, the following should be noted:
1. Authors' review of literature selected randomly from the PubMed, Scopus and Google Scholar databases cannot be considered satisfactory. In my opinion, a complete review of the literature on this topic published over the past 5 years in publications indexed in PubMed, Scopus and Web of Science would be more informative and would more fully reflect the current state of the problem.
2. Quote: "Studies show that patellofemoral pressure is 5 to 7 times more of the body weight when standing up, 2 to 3 time more than the body weight when climbing stairs and 20 times more when jumping [4, 5]".
Question: How you can compare such concepts as “patellofemoral pressure” and “body weight”, which have different dimensions?
The article can be recommended for publication in the journal after answering above questions.
Author Response
(1) "Authors' review of literature selected randomly from the PubMed, Scopus and Google Scholar databases cannot be considered satisfactory. In my opinion, a complete review of the literature on this topic published over the past 5 years in publications indexed in PubMed, Scopus and Web of Science would be more informative and would more fully reflect the current state of the problem."
- The literature review was conducted as a complement to the case presentation, aiming to objectify our observations in line with the literature data. A full literature review or meta-analysis was not an objective of our study. We did not limit our search to studies published in the last 5 years because we wanted to highlight changes in surgeons' opinions on this topic over the years. Initially it was a method not taken into account, then it became a widely used method, and at present it represents a great controversy among orthopaedic surgeons.
(2) "Question: How you can compare such concepts as “patellofemoral pressure” and “body weight”, which have different dimensions?"
- Biomechanical studies in orthopaedics assess intra-articular pressure, kg force, in relation to the weight of the person's body, for a better understanding of the forces that occur in that joint. These data are taken from published and approved studies.